# Structural Change Analysis of Cerianite in Weathered Residual Rare Earth Ore by Mechanochemical Reduction Using X-Ray Absorption Fine Structure

**Tatsuya Kato [1], Yuki Tsunazawa [2] 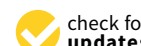, Wenying Liu [3] and Chiharu Tokoro [4,\*]**

[1] Graduate School of Creative Science and Engineering, Waseda University, 3-4-1 Okubo, Shinjuku-ku, Tokyo 169-8555, Japan; tatsuya.kato@aoni.waseda.jp

[2] National Institute of Advanced Industrial Science and Technology, 1-1-1 Higashi, Tsukuba, Ibaraki 305-8567, Japan; tsunazawa-y@aist.go.jp

[3] Department of Materials Engineering, University of British Columbia, 309-6350 Stores Road, Vancouver, BC V6T 1Z4, Canada; wenying.liu@ubc.ca

[4] Faculty of Science and Engineering, Waseda University, 3-4-1 Okubo, Shinjuku-ku, Tokyo 169-8555, Japan

\* Correspondence: tokoro@waseda.jp; Tel.: +81-3-5286-3320

**Abstract:** Prolonged high-intensity grinding can modify the crystal structure of solid substances and/or induce chemical reaction, which is referred to as mechanochemical reaction. Such reactions can exert positive influences on hydrometallurgical processes, therefore, many researchers have applied mechanochemical reactions for metals dissolution from minerals. The mechanism of mechanochemical reaction has been investigated using solid analyses and simulations. Structural changes caused by mechanochemical reactions are not yet sufficiently clarified because the ground samples are amorphous. The objective of this study was to analyze structural changes of cerianite in weathered residual rare earth ore by mechanochemical reduction. The ore was ground by planetary ball milling for 10, 60 and 720 min. Structural change was analyzed by the X-ray absorption near-edge structure and extended x-ray absorption fine structure analysis at the cerium $L_{III}$- and K-edges. These analyses revealed that the structural change of cerianite in this ore induced by mechanochemical reduction involved oxygen vacancy production. The process of the oxygen vacancy formation was closely coupled with the quantum effect of localization–delocalization of the 4f electron of cerium.

**Keywords:** oxygen vacancy; extended X-ray absorption fine structure; high-intensity grinding; local structure; quantum effect

## 1. Introduction

Cerianite is tetravalent cerium dioxide ($CeO_2$) and has the fluorite structure (space group ($Fm\bar{3}m$)) [1]. In nature, cerianite exists in rare earth ores [2–5]. Cerianite has oxygen storage capacity due to a facile redox reaction from tetra- to trivalent cerium, as described in Equation (1) [6], and the ability to improve dispersion of noble metals [7] and provide thermal stabilization of alumina supports [8]. Owing to these unique properties, cerianite has been widely used as a promoter of the three-way catalyst for automobile exhaust emission control [6,7].

$$CeO_2 \rightarrow CeO_{2-x} + \frac{x}{2}O_2 \tag{1}$$

Since this redox reaction is involved in the oxygen, many researchers investigated the influence of oxygen partial pressure in the experimental system [9,10]. Zinkevich et al. [9] and Panlener et al. [10] reported

that the redox reaction of cerianite was promoted in the lower oxygen partial pressure. In addition, it has been reported that the reducibility of cerianite is enhanced by the addition of yttrium [11,12], gadolinium [13,14], and zirconium to form a solid solution [15,16]. Previous papers [11–16] reported that a solid solution of cerianite was reduced from tetra- to trivalent cerium at moderate temperatures (600–700 °C), while pure cerianite reduced only at high temperatures (<1200 °C) [17]. The reaction properties, such as structural changes and activation energy of solid solutions of cerianite, have been investigated using both experimental and simulation approaches. In the experimental approaches, the X-ray diffraction analysis [18], electrical conductivity measurements [19], and X-ray absorption fine structure (XAFS) analysis [11–16] were usually used to investigate the reaction mechanism and structural changes induced by the addition of yttrium, gadolinium, and zirconium. In the simulation approaches, the activation energy for oxygen migration in a solid solution of cerianite was calculated, using the structural change analysis based on XAFS [20,21].

It is known that the redox reaction of cerianite, as described by Equation (1), only occurs when heating above moderate temperatures; however, our previous paper [2] recently reported that this redox reaction of cerianite in a weathered residual rare earth ore also occurred by planetary ball milling at room temperature.

A reaction that occurs on grinding, such as this redox reaction, is generally known as a mechanochemical reaction [22,23]. High-intensity grinding, like planetary ball milling, has the ability to modify crystal structures of solid substances and/or induce a chemical reaction, which is referred to as mechanochemical reaction [24,25]. Mechanochemical reactions have a positive influence on hydrometallurgical processes and have been applied for metals dissolution from ores [2,5,22,23,26–28]; however, these studies only focused on investigation of the reaction mechanism. There is still limited knowledge of mechanochemical reactions and their theory has not been established. To apply mechanochemical reactions with high-intensity grinding to practical use in hydrometallurgical processes, it is strongly desired to establish the theory.

The objective of this study was to clarify the structural change of cerianite in a weathered residual rare earth ore during mechanochemical reduction. We performed grinding on the ore using a planetary ball mill. The ground samples were analyzed by XAFS at the cerium $L_{III}$- [2] and K-edges because that was amorphous and the concentration of cerium in the ore was much lower than that of other elements. We then performed extended the x-ray absorption fine structure (EXAFS) analysis using the XAFS spectra at the cerium K-edge to clarify the structural change of cerianite in the ore caused by the planetary ball milling.

## 2. Materials and Methods

### 2.1. Analytical Sample

The weathered residual rare earth ore used in this study was obtained from the Republic of South Africa. Detailed characterizations were described in our previous paper [2]. Briefly, as shown in Tables 1 and 2, the major chemical composition of this ore was determined by the x-ray fluorescence (XRF; ZSX Primus III+, Rigaku Corporation, Tokyo, Japan), while concentrations of the rare earth elements were analyzed by the laser-ablation inductively coupled plasma mass spectrometry (LA–ICP–MS; 7500 Series, Agilent Technologies, Santa Clara, CA, USA). The mineral liberation analysis (MLA; QuantaF Co., FEI, Hillsboro, OR, USA) revealed that this ore contained four types of cerium minerals: Britholite-(Ce) [$(Ce,Ca,Th,La, Nd)_5(SiO_4,PO_4)_3(OH,F)$], cerianite [$(Ce,Th)O_2$], cerite [$Ce_9Fe(SiO_4)_6[(SiO_3),(OH)](OH)_3$], and monazite-(Nd) [$(Nd,Ce,La)(P,Si)O_4$]. Of these, britholite-(Ce), cerite, and monazite-(Nd) were trivalent cerium minerals, while cerianite was the only tetravalent cerium mineral.

Grinding experiments using a planetary ball mill (PM100, Verder Scientific Co. Ltd., Tokyo, Japan) were performed under the same conditions as reported in our previous paper [2]. Briefly, each experiment employed 100 g ore; the rotation speed of the mill was fixed at 300 rpm; 25 chromium steel

balls of 19 mm diameter were used as the grinding media. Grinding time was 10, 60 and 720 min. The specific surface areas of the samples, without and with grinding, as measured by a high-efficiency specific surface area and pore distribution analyzer (ASAP 2020 Series, Shimadzu, Kyoto, Japan), are shown in Table 3.

XAFS analysis at the cerium $L_{III}$-edge of samples with and without grinding was performed using the BL5S1 beamline at Aichi Synchrotron Radiation Center, Seto, Japan. XAFS spectra at the cerium $L_{III}$-edge have an advantage in evaluating the oxidation state of cerium. The valence of cerium was evaluated by Kyoto the X-ray absorption near-edge structure (XANES) analysis in the range of 5720–5740 eV, using cerium fluoride ($CeF_3$) and tetravalent cerium oxide ($CeO_2$) as reference materials. Briefly, the method of XANES analysis is to represent by superposition the peak intensity of samples using that of reference materials. It should be noted that the calculated results by the XANES analysis have a margin of error about 10%. The concentrations of tri- and tetravalent cerium in the samples with and without grinding are shown in Table 4, which was discussed in our previous paper [2].

**Table 1.** Chemical composition of weathered residual rare earth ore, as analyzed by X-ray fluorescence (mass %) [2].

| $SiO_2$ | $Fe_2O_3$ | $Al_2O_3$ | $Na_2O$ | MgO | $P_2O_5$ | $K_2O$ | CaO | $TiO_2$ | MnO | Others |
|---------|-----------|-----------|---------|-----|----------|--------|-----|---------|-----|--------|
| 68.6 | 16.2 | 7.3 | 0.3 | 0.4 | 0.3 | 2.1 | 0.2 | 0.3 | 0.2 | 4.1 |

**Table 2.** Concentrations of rare earth elements in weathered residual rare earth ore, as analyzed by laser-ablation inductively coupled plasma mass spectrometry ($mg/dm^3$) [2].

| Sc | Y | La | Ce | Pr | Nd | Sm | Eu |
|----|-----|-----|------|-----|------|------|------|
| 5.3 | 1384.1 | 926.5 | 2495.5 | 236.8 | 903.3 | 200.0 | 11.2 |
| Gd | Tb | Dy | Ho | Er | Tm | Yb | Lu |
| 224.2 | 37.4 | 236.2 | 50.4 | 153.7 | 23.1 | 137.9 | 19.6 |

**Table 3.** Specific surface area of sample with and without grinding [2].

| Grinding Time (min) | Specific Surface Area ($m^2/g$) |
|---------------------|----------------------------------|
| 0 | 16.2 |
| 10 | 23.8 |
| 60 | 19.1 |
| 720 | 10.4 |

**Table 4.** Concentrations of tri- and tetravalent cerium in samples with and without grinding, based on X-ray absorption near-edge structure analysis at the cerium $L_{III}$-edge (%) [2].

| Grinding Time (min) | Concentration of Trivalent Cerium | Concentration of Tetravalent Cerium |
|---------------------|-----------------------------------|-------------------------------------|
| 0 | 50.71 | 49.29 |
| 10 | 54.47 | 45.53 |
| 60 | 70.20 | 29.80 |
| 720 | 89.80 | 10.20 |

## 2.2. X-Ray Absorption Fine Structure Analysis at Cerium K-edge

The XAFS analysis, including XANES and EXAFS, is a very powerful analytical technique for an amorphous sample. In particular, the local structure in a specific atom can be analyzed using EXAFS. Low-to-medium-energy synchrotron facilities are relatively accessible, so most of the XAFS analysis for cerianite has been conducted at the cerium $L_{III}$-edge ($5.7 \times 10^3$ eV) [29–31]. The EXAFS analysis at the cerium $L_{III}$-edge has a somewhat lower accuracy for derived structural parameters due to its limited data range, arising from the small energy separation between the $L_{III}$- and $L_{II}$-edges [29–31].

Therefore, the XAFS analysis for cerianite was conducted at the cerium K-edge ($4.0 \times 10^4$ eV). This way, a more accurate local structure around the cerium atom could be obtained because of the larger EXAFS data range, as described previously [32].

To analyze the structural change caused by grinding, the XAFS analysis at the cerium K-edge was conducted on samples with and without grinding using the BL14B2 beamline at the SPring-8 Synchrotron Radiation Facility, Japan. All spectra were obtained at room temperature. The electron storage ring was operated at 8.0 GeV with a stored current of 99.6 mA. Energy was scanned from 40,290 eV to 41,233 eV using a step size of 6 eV. Continuous X-rays from synchrotron radiation were monochromatized using a silica (311) double-crystal monochromator. The XAFS spectra for all samples were obtained in fluorescence mode using a 19-element germanium (Ge) solid-state detector. Cerium phosphate ($CePO_4$) and $CeO_2$ were used as reference materials.

The EXAFS functions were derived from the raw XAFS spectra by the pre- and post-edge linear background subtraction and then normalized with respect to the edge jump. After being $k^3$-weighted, where $k$ is the photoelectron wave number, the EXAFS function was Fourier transformed from the $k^3$-weighted EXAFS function to a radial distribution function (RDF) using a Hanning window function within $1–12 \times 10^{10}$ m$^{-1}$. Structural parameters for different coordination shells surrounding both the central tri- and tetravalent cerium atoms, i.e., coordination number, atomic distance, and Debye–Waller factor, were obtained by curve fitting using both the $k^3$-weighted EXAFS function and RDF. Structural parameters were obtained from fitting the RDF in the interval of $1.5–4.0 \times 10^{-10}$ m, using contributions from the first to fourth coordination shells (Ce(IV)–O, Ce(III)–O, Ce(III)–Ce(III), and Ce(IV)–Ce(IV)). The theoretical phases and amplitude functions for these shells were calculated using the FEFF 6.0 software [33,34]. All EXAFS analyses were performed using the Athena and Artemis software [35].

## 3. Results and Discussion

### 3.1. X-Ray Absorption Near-Edge Structure Analysis

The XAFS spectra at the cerium K- and L$_{III}$-edges [2] corresponding to the XANES region are shown in Figure 1 for samples with and without grinding. Tri- and tetravalent cerium peaks at the cerium K-edge XAFS spectra were observed at 40,460 eV and 40,470 eV, respectively. The valence of cerium was already identified by the XANES analysis at the cerium L$_{III}$-edge XAFS spectra, as shown in Table 4 [2]. In addition, the specific surface area was already measured by a high-efficiency specific surface area and pore distribution analyzer, as shown in Table 3 [2]. The concentration of trivalent cerium significantly increased with grinding time, especially after 10 min of grinding. On the other hand, the specific surface area shown in Table 3 increased by 10 min of grinding, but decreased after 10 min of grinding. Thus, it was suggested that the planetary ball milling energy to generate a new surface area by 10 min of grinding, but occur mechanochemical reaction as described in Equation (1) after 10 min grinding [2]. It was confirmed by both the cerium K- and L$_{III}$-edge XAFS spectra that the energetic shift corresponded to the change in valence of cerium (Figure 1a). The results of the XANES analysis using the XAFS spectra at the cerium K-edge are shown in Table 5. Compared with the data in Table 4, the trend, in which the concentration of trivalent cerium significantly increased with grinding time, was the same. These results showed that the XAFS spectra at the cerium K-edge were sufficient to identify the valence of cerium in samples both with and without grinding [29,30].

Cerium is the first element in the Periodic Table with a partially occuped f orbital: Its arrangement of electrons is [Xe core]$4f^1$[Xe core]$5d^16s^2$ ($1s^22s^22p^63s^23p^63d^{10}4s^24p^64d^{10}4f^15s^25p^65d^16s^2$). From Figure 1b, the XAFS spectra at the cerium L$_{III}$-edge of $CeO_2$ had two characteristic peaks at 5730 and 5737 eV, while that of $CeF_3$ had only one peak at 5725 eV. The peaks of $CeO_2$ at 5730 eV and 5737 eV are derived from the 2p to $(4f^1)$5d electronic transition, which results from interactions between the cerium 4f electron and the nearest oxygen 2p electron, and from the 2p to $(4f^0)$5d electronic transition, respectively [36,37]. The peak of $CeF_3$ at 5725 eV is derived from the 2p to $(4f^1)$5d electronic transition [36,37]. In $CeO_2$, all tetravalent cerium normally leaves the host atoms

and transfers into the 2p bands of two oxygen atoms; in trivalent cerium trioxide ($Ce_2O_3$), the cerium 4f electron is fully localized [38]. From the above discussion, it is suggested that the structural change of cerianite in weathered residual rare earth ore during mechanochemical reduction involved localization–delocalization of the cerium 4f electron.

**Table 5.** Concentrations of tri- and tetravalent cerium in samples with and without grinding, based on X-ray absorption near-edge structure analysis at cerium K-edge (%).

| Grinding Time (min) | Concentration of Trivalent Cerium | Concentration of Tetravalent Cerium |
|---|---|---|
| 0 | 30.46 | 69.54 |
| 10 | 31.92 | 68.08 |
| 60 | 50.01 | 49.99 |
| 720 | 74.38 | 25.62 |

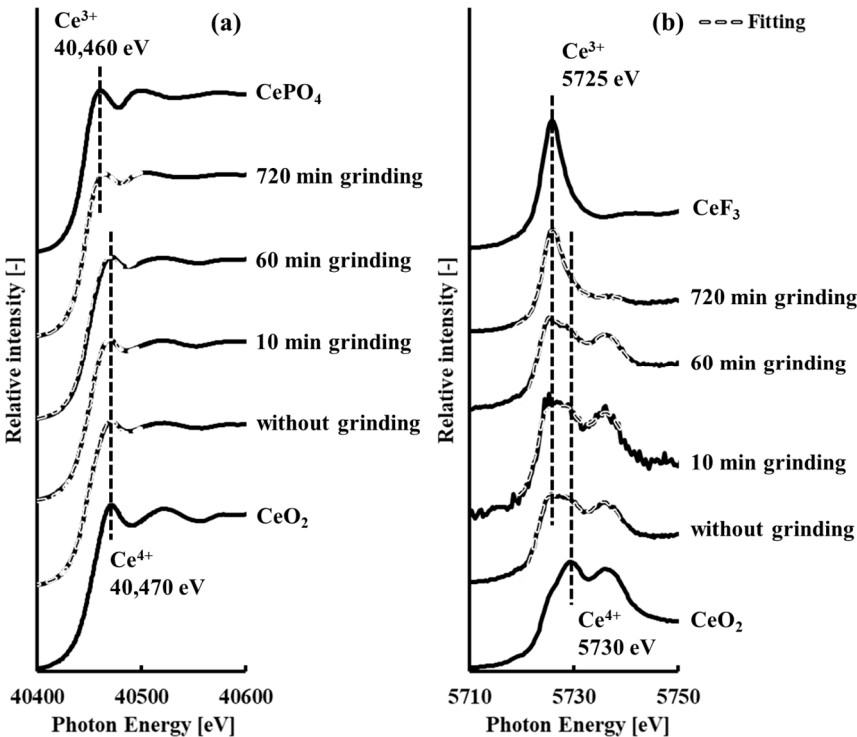

**Figure 1.** Cerium (**a**) K- and (**b**) $L_{III}$-edge X-ray absorption fine structure spectra [2] corresponding to X-ray absorption near the edge structure region of samples with and without grinding, showing resolved components of tri- and tetravalent cerium based on cerium fluoride or cerium phosphate and tetravalent cerium oxide reference spectra, respectively.

*3.2. Extended X-Ray Absorption Fine Structure Analysis*

The $k^3$-weighted EXAFS spectra and RDF of $CeO_2$ of samples with and without grinding are shown in Figures 2 and 3, respectively. The curve-fitted results are plotted and the corresponding parametric values are listed in Table 6. It should be noted that the third shell (Ce(III)–Ce(III)) contribution was quite minor relative to that of the fourth shell (Ce(IV)–Ce(IV)), and a significant overlap existed between these two shells in the RDF.

Figure 2 shows that the $k^3$-weighted EXAFS spectra of the sample without grinding was similar to that of the sample ground for 10 min, but these differed from those of samples ground for 60 min and 720 min. This trend is also shown in the RDF (Figure 3). These results clearly showed that the structure of cerianite in the weathered residual rare earth ore started to change after 10 min of grinding.

The curve-fitting results in Table 6 show that the coordination numbers of Ce(IV)–O and Ce(IV)–Ce(IV) in the cerianite decreased as grinding time increased, while those of Ce(III)–O and

Ce(III)–Ce(III) increased. From the above discussion, the results of the EXAFS analysis suggested that the structure of the ore changed from cerianite to $Ce_2O_3$ because of a mechanochemical reduction [2]. Cerianite has the fluorite structure (space group ($Fm\bar{3}m$)) [1] (Figure 4a). Skorodumova et al. [38] reported that the C-type structure of $Ce_2O_3$ (space group ($Ia\bar{3}$)) was produced on completion of the reduction process [39,40] (Figure 4b). This structure (Figure 4b) can be constructed from eight unit cells of cerianite (Figure 4a) with 25% oxygen vacancies ordered in a particular way [38]. The structure change from cerianite to the C-type structure of $Ce_2O_3$ involves minimal reorganization of the skeleton arrangement of the cerium atoms [38]. This structural property should facilitate the excellent reversibility of the reduction–oxidation process.

The atomic distances of both Ce(IV)–O and Ce(III)–O in cerianite also decreased as grinding time increased, as shown in Table 6. This trend was similar to the case of $Gd_2O_3$-doped cerianite [13]: Ohashi et al. reported that when gadolinium was doped at 30% in cerianite, the Ce(IV)–O and Gd(III)–O atomic distances decreased by $0.032 \times 10^{-10}$ m and $0.024 \times 10^{-10}$ m, respectively [13]. Compared with this result, the decrease of atomic distance in the present system was large; thus, the decrease of atomic distances of Ce(IV)–O and Ce(III)–O in the cerianite represented significant structural changes.

The decreases of the Ce(IV)–O and Ce(III)–O atomic distances in the cerianite showed that oxygen vacancies occurred in the structure on grinding, and that oxygen ions surrounding these vacancies around both tri- and tetravalent cerium ions relaxed toward their adjacent vacancies [21]. Skorodumova et al. [38] reported that it required $4.4 \times 10^2$ kJ/mol to form an oxygen vacancy in pure cerianite, based on the full-potential linear muffin tin orbitals generalized gradient approximation (FP–LMTO–GGA) [41,42]. In our experimental system, it is suggested that this energy was mainly provided by the planetary ball milling. In addition, it is suggested that the distribution of oxygen vacancies in the stucture of cerianite is related to the valences and ionic radii of tri- and tetravalent cerium ions. In this system, the ionic radii of eight-coordinated tri- and tetravalent cerium are $1.143 \times 10^{-10}$ m and $0.97 \times 10^{-10}$ m, respectively [43]. These ionic radii are relatively similar, so oxygen vacancies are considered to be favored near trivalent cerium ions to ensure electrical neutrality. Indeed, Skorodumova et al. [38] reported that it required only 25 kJ/mol, which was the lowest energy among positions that could form an oxygen vacancy, to form an oxygen vacancy adjacent to trivalent cerium in pure cerianite.

Formation of a cluster composed of oxygen vacancies and trivalent cerium ions has been proposed by some researchers, using the cluster model for the structure of cerianite, which is specified by the number of oxygen vacancies and their configuration and those of cations around the vacancies [13,20,21]. According to the cluster model, two models expressing the local structures around oxygen vacancies are proposed, as shown in Figure 5: One oxygen vacancy in the structure of cerianite is introduced for every two trivalent cerium ions, as required by the electroneutrality condition. In the type A model, one unit cell of cerianite contains one trivalent cerium and one oxygen vacancy, while the other unit cell contains one trivalent cerium and no oxygen vacancy; in the type B model, one unit cell of cerianite contains two trivalent ceriums and one oxygen vacancy. If only type A were formed, the Ce(III)–O atomic distance in the cerianite would not be expected to decrease as the concentration of trivalent cerium in the ore increased with grinding time; if only type B were formed, a similar trend would occur. If both types A and B co-existed, and the proportion of type A decreased and that of type B increased with grinding, then the Ce(III)–O atomic distance in the cerianite would decrease as the concentration of trivalent cerium in the ore increased with grinding time. From the above discussion, it is suggested that oxygen vacancies occurred in the cerianite in the weathered residual rare earth ore and changed the structure from that of fluorite to one in which types A and B co-existed and the proportion of type B increased as grinding time by planetary ball milling increased.

From the results of the XANES and EXAFS analysis, it is suggested that the process of oxygen vacancy formation was closely coupled with the quantum effect of localization–delocalization of the 4f electron of cerium. Oxygen in cerianite has two extra electrons in the p band, provided by tetravalent cerium. When an oxygen vacancy is produced in cerianite, it is suggested that these two electrons

are left behind and occupy the lowest possible empty state, which is the f band of cerium. Thus, the tetravalent cerium in the cerianite is reduced to trivalent cerium (Figure 6).

**Table 6.** Cerium fitting results of radial distribution function for samples with and without grinding, assuming four kinds of cerium shells for tetravalent cerium oxide.

| Sample | Shell | CN | R ($\times$ m$^{-10}$) | $\sigma^2$ ($\times$ m$^{-20}$) | $\Delta E_0$ |
|---|---|---|---|---|---|
| CeO$_2$ | Ce(IV)–O | 7.7 | 2.35 | 0.009 | −10.3 |
| | Ce(III)–O | | - | | |
| | Ce(III)–Ce(III) | | - | | |
| | Ce(IV)–Ce(IV) | 12.2 | 3.84 | 0.005 | |
| Without grinding | Ce(IV)–O | 8.7 | 2.35 | 0.02 | −9.3 |
| | Ce(III)–O | 0.2 | 2.46 | 0.04 | |
| | Ce(III)–Ce(III) | 0.3 | 3.81 | 0.002 | |
| | Ce(IV)–Ce(IV) | 12.1 | 3.83 | 0.008 | |
| 10 min grinding | Ce(IV)–O | 8.0 | 2.36 | 0.02 | −10.6 |
| | Ce(III)–O | 0.06 | 2.47 | −0.01 | |
| | Ce(III)–Ce(III) | 0.2 | 3.81 | 0.001 | |
| | Ce(IV)–Ce(IV) | 11.4 | 3.84 | 0.008 | |
| 60 min grinding | Ce(IV)–O | 4.0 | 2.31 | 0.006 | 2.4 |
| | Ce(III)–O | 3.9 | 2.42 | 0.02 | |
| | Ce(III)–Ce(III) | 7.4 | 3.77 | 0.008 | |
| | Ce(IV)–Ce(IV) | 5.4 | 3.79 | 0.009 | |
| 720 min grinding | Ce(IV)–O | 0.1 | 2.30 | 0.009 | −3.4 |
| | Ce(III)–O | 8.4 | 2.42 | 0.03 | |
| | Ce(III)–Ce(III) | 10.0 | 3.76 | 0.04 | |
| | Ce(IV)–Ce(IV) | 2.6 | 3.79 | 0.0009 | |

CN: coordination number; R ($\times$ m$^{-10}$): atomic distance; $\Delta E_0$: threshold E$_0$ shift; $\sigma^2$ ($\times$ m$^{-20}$): Debye–Waller factor.

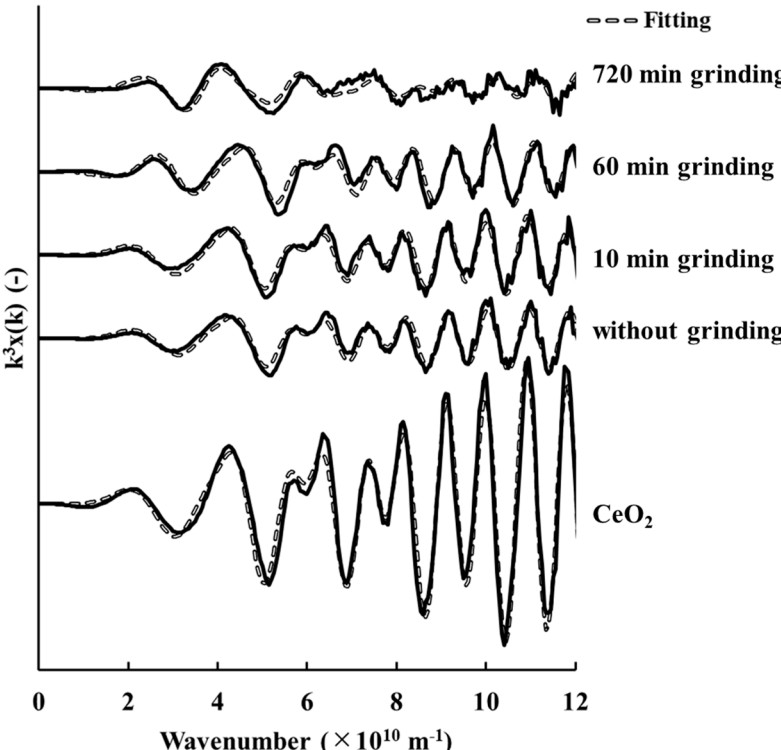

**Figure 2.** $k^3$-weighted cerium K-edge extended X-ray absorption fine structure spectra of tetravalent cerium oxide for samples with and without grinding.

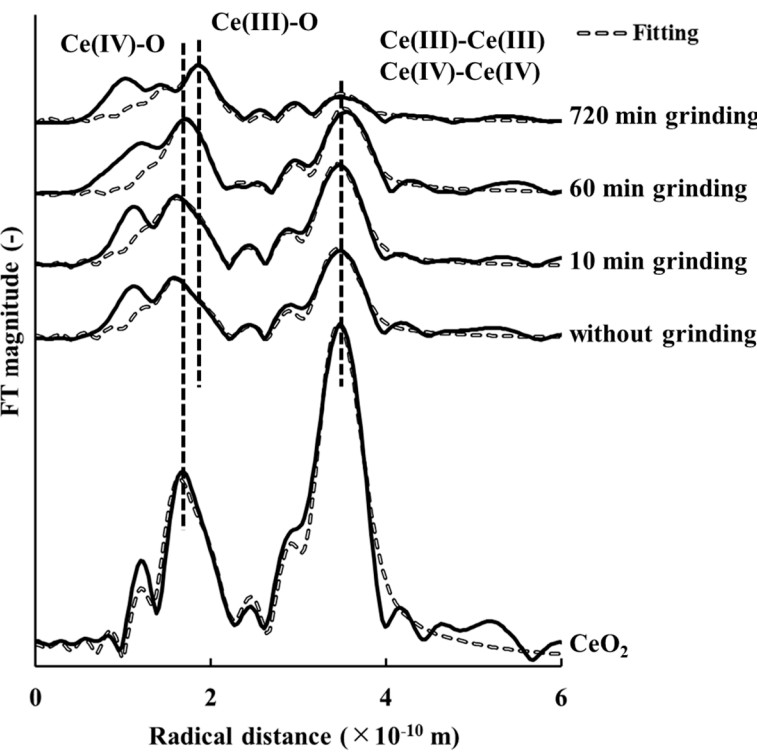

**Figure 3.** Radial distribution functions of tetravalent cerium oxide for samples with and without grinding.

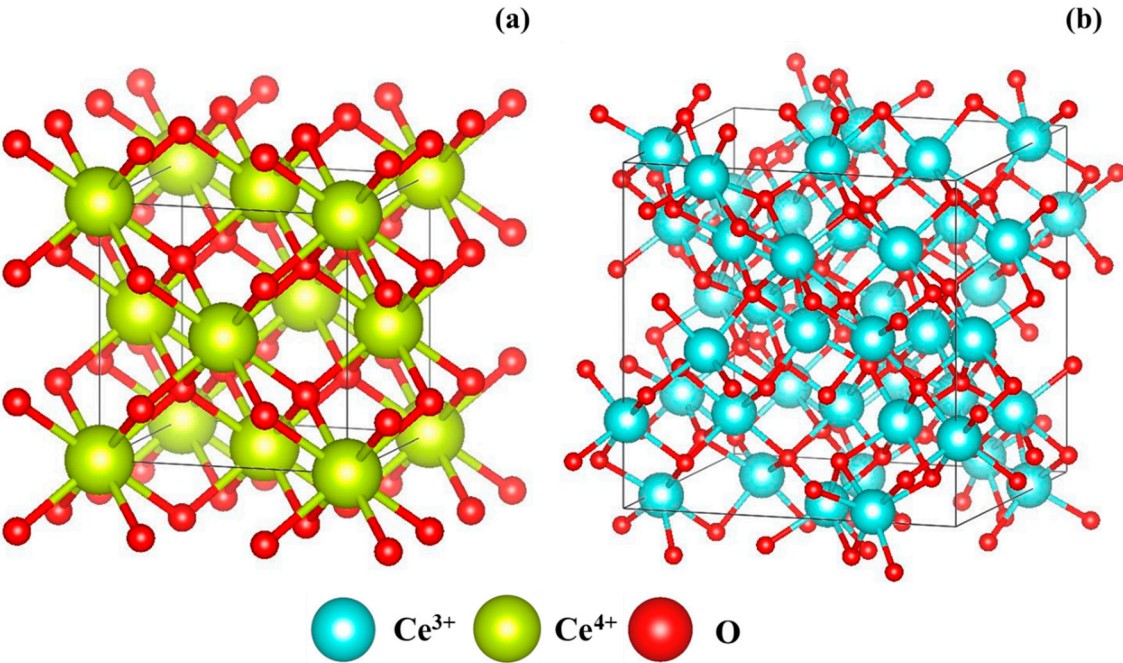

**Figure 4.** Lattice unit cells for (**a**) tetravalent cerium oxide and (**b**) C-type structure of trivalent cerium [44].

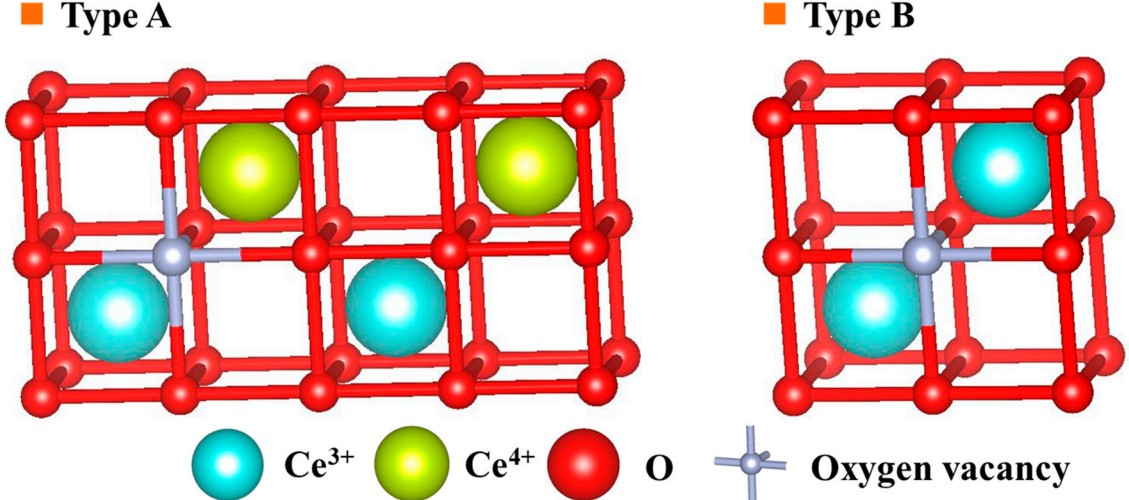

**Figure 5.** Two models expressing the local structure around oxygen vacancies in tetravalent cerium oxide [44].

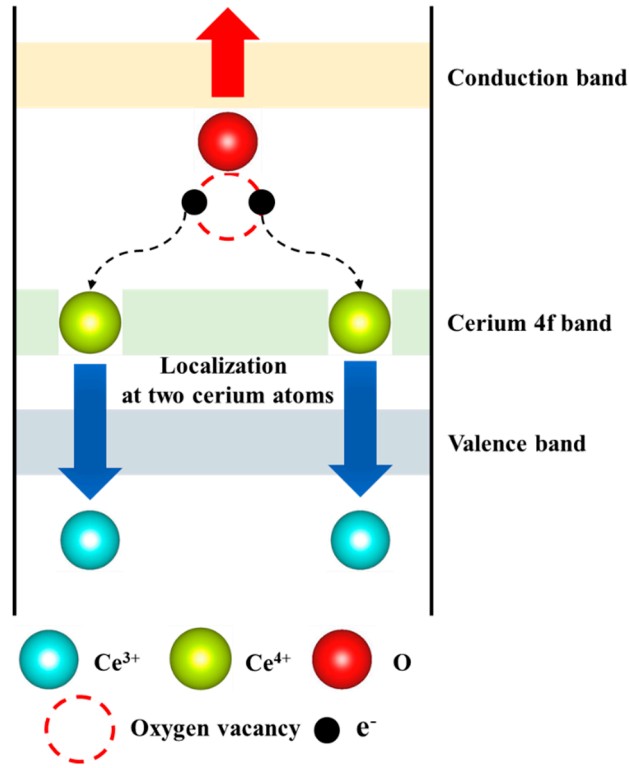

**Figure 6.** Schematic process of oxygen vacancy formation in tetravalent cerium oxide.

## 4. Conclusions

The objective of this study was to clarify the structural change of cerianite in weathered residual rare earth ore during mechanochemical reduction. Results of the XANES and EXAFS analysis revealed that the structural change involved oxygen vacancies produced in the cerianite. The process of the oxygen vacancy formation was closely coupled with the quantum effect of localization–delocalization of the 4f electron of cerium. The EXAFS analysis showed that the atomic distances of both Ce(III)–O and Ce(IV)–O decreased, which indicated that the cerianite changed from the fluorite structure to one in which types A and B co-existed and the proportion of type B increased as grinding time by planetary ball milling increased.

**Author Contributions:** Conceptualization, C.T., W.L. and T.K.; Formal analysis, T.K.; Investigation, T.K. and Y.T.; Resources, Y.T.; Writing—original draft preparation, T.K.; Writing—review and editing, Y.T., W.L. and C.T.; Supervision, C.T.; Project administration, C.T.; Funding acquisition, C.T.

**Funding:** This study was partially supported by the Hosokawa Powder Technology Foundation, under Grant Nos. 18504 and 2018, and the Joint Research Center for Environmentally Conscious Technologies in Materials Science (Project No. 30006) at ZAIKEN, Waseda University.

**Acknowledgments:** The synchrotron radiation experiments were performed using a BL5S1 beamline courtesy of the Aichi Synchrotron Radiation Center, Aichi Science & Technology Foundation, Aichi, Japan (Proposal No. 201801021), and BL14B2 beamline of SPring-8, with approval of the Japan Synchrotron Radiation Research Institute (Proposal No. 2018A1696). Part of this work was performed as a component of the activities of the Research Institute of the Sustainable Future Society, Waseda Research Institute for Science and Engineering, Waseda University, and the Joint Usage/Research Center on Joining and Welding, Osaka University. We thank Kathryn Sole from Edanz Group (www.edanzediting.com/ac) for editing a draft of this manuscript.

**Conflicts of Interest:** The authors declare no conflict of interest.

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
