# Peer review of "Structural Change Analysis of Cerianite in Weathered Residual Rare Earth Ore by Mechanochemical Reduction Using X-Ray Absorption Fine Structure"

_minerals, doi:10.3390/min9050267_

Round 1
Reviewer 1 Report
This paper discusses on the structural change of cerianite with XAFS. The results are interesting.
The structural change would occur during high energy ball milling and the atmosphere in the vial. Have you controlled the atmosphere in the vial? Oxygen in the vial might be related to the structural change of cerianite. If possible, discussion on the influence of atmosphere should be shown in the text.
Line 161, localication should be localization.
Table 5 30.46+69.55 is not 100.00. Those values could be something wrong.
Author Response
We appreciate the reviewer’s comment/suggestion. Please kindly find with the attached point-by-point response.

Reviewer 2 Report
The paper entitled “Structural change analysis of cerianite in weathered residual rare earth ore by mechanochemical reduction using x-ray absorption fine structure” is focused on the influence of grinding time (10, 60 and 720 min.) on the mechano-chemical treatment of cerianite.
Broad comments
The topic addressed in the manuscript is very interesting, the aims of the paper were well described in the introduction section, and the result are well reported. However, many data have already been published in:
Kato, T.; Granata, G.; Tsunazawa, Y.; Takagi, T.; Tokoro, C. Mechanism and kinetics of enhancement of cerium dissolution from weathered residual rare earth ore by planetary ball milling. Minerals Engineering 2019, 134, 365 – 371
Table 1 is the same as table 1 of the following paper: Kato, T.; Granata, G.; Tsunazawa, Y.; Takagi, T.; Tokoro, C. Mechanism and kinetics of enhancement of cerium dissolution from weathered residual rare earth ore by planetary ball milling. Minerals Engineering 2019, 134, 365 – 371
Table 2 is the same as table 2 of the following paper: Kato, T.; Granata, G.; Tsunazawa, Y.; Takagi, T.; Tokoro, C. Mechanism and kinetics of enhancement of cerium dissolution from weathered residual rare earth ore by planetary ball milling. Minerals Engineering 2019, 134, 365 – 371
Table 3 reports the same data as in Figure 1 of the following paper: Kato, T.; Granata, G.; Tsunazawa, Y.; Takagi, T.; Tokoro, C. Mechanism and kinetics of enhancement of cerium dissolution from weathered residual rare earth ore by planetary ball milling. Minerals Engineering 2019, 134, 365 – 371
Table 4 is the same as table 3 of the following paper: Kato, T.; Granata, G.; Tsunazawa, Y.; Takagi, T.; Tokoro, C. Mechanism and kinetics of enhancement of cerium dissolution from weathered residual rare earth ore by planetary ball milling. Minerals Engineering 2019, 134, 365 – 371.
(this reported by the authors, line 98)
Figure 1b is similar as Fig. 2 of the following paper: Kato, T.; Granata, G.; Tsunazawa, Y.; Takagi, T.; Tokoro, C. Mechanism and kinetics of enhancement of cerium dissolution from weathered residual rare earth ore by planetary ball milling. Minerals Engineering 2019, 134, 365 – 371
Grinding time (i.e., 10, 60 and 720 min.) should appear either in the abstract and/or in the introduction.
Have you evaluated the temperature increase of the grinding container with the milling?
Table 3 (Specific surface area of sample without and with grinding) was not commented on throughout the paper.
Details,
Pag. 2 line 51-56 a lot of recent papers focus on grinding and should be cited:
-Bloise, A.; Catalano, M.; Gualtieri, A.F. Effect of Grinding on Chrysotile, Amosite and Crocidolite and Implications for Thermal Treatment. Minerals 2018, 8, 135.
-Bloise, A.; Kusiorowski, R.; Gualtieri, A.F. The Effect of Grinding on Tremolite Asbestos and Anthophyllite Asbestos. Minerals 2018, 8, 274.
Pag. 5 line 185-186 Skorodumova et al.[?], please add the reference number after et al., (and also everywhere in the paper).
Author Response

(The authors gave the same response as above.)

Reviewer 3 Report
The manuscript is well written, in my opinion. It shows the applicability of XAFS for the study of unknown amorphous states within mechanochemical reactions.
I have though some minor comments:
- Table1, 2, 3, 4, and 5: please provide an uncertainty to the values you calculate.
- Line 98: Please describe shortly the method you used in your previous paper [ref 2] for the calculation.
- Line 112/113. Change to: "(...) separation between the LIII-LII-edges [25-27]. Therefore, XAFS analysis (...) K-edge (4.0x10^4 eV). This way, more accurate local structure (...)."
- Line 121: why did you use a Si (311)? Yu have extremely high energy for scans you did with 6 eV steps. And, you have lower photon flux...
- Paragraph 3.1 (lines137 to 148): I don´t understand the need to perform XANES scans @ the Ce-K edge. For valence detection, it is more appropriate to use the LIII-edge, as you already did. Why did you us the K-edge for this purpose? Furthermore, the XANES curves you show in figure 1a reveal an edge shift of 10 eV. At this energy, the core-hole broadening is about 15 eV (Grant Bunker, XAFS - an introduction). So I wouldn´t withdraw much interpretation out of this shift you see.
For the EXAFS part I see the need. As you already explained, and correctly, in order to don´t have overlap of other absorption lines.

Author Response

(The authors gave the same response as above.)

Round 2
Reviewer 2 Report
Dear Authors,
congratulations for the interesting work.
Best Regards